

# Technical Note: Bimodality in Mesospheric OH Rotational Population Distributions and Implications for Temperature Measurements

Konstantinos S. Kalogerakis[1]

[1]Center for Geospace Studies, SRI International, Menlo Park, California, USA

*Correspondence to*: Konstantinos S. Kalogerakis (ksk@sri.com)

**Abstract.** Emission from the OH Meinel bands is routinely used to determine rotational temperatures that are considered proxies for the kinetic temperature near the mesopause region. Previous observations determined OH rotational temperatures that show a dependence on the vibrational level, with the temperature rising overall as the OH vibrational quantum number v

increases. The source of this trend is not well understood and has generally been attributed to deviations from thermodynamic equilibrium. This Technical Note demonstrates that the existence of bimodal OH rotational population distributions is an inherent feature of rotational relaxation in gases and can provide an explanation for the previously reported temperature trend. The use of only a few lines from rotational transitions involving low rotational quantum numbers to determine rotational temperatures does not account for the bimodality of the OH rotational population distributions and

leads to systematic errors overestimating the OH rotational temperature. This Note presents selected examples, discusses the relevant implications, and considers strategies that could lead to more reliable OH rotational temperature determination.

## 1 Introduction

The hydroxyl radical is an important species in the middle atmosphere of the Earth. At altitudes around 87 km, the exothermic reaction of ozone with atomic hydrogen produces rotationally and vibrationally excited hydroxyl, OH(v), in

vibrational levels v = 5–9 [Adler-Golden, 1997; Khomich et al., 2008; von Savigny 2017; and references therein]. The radiative decay of OH(v) in the visible and infrared region of the electromagnetic spectrum, known as the OH Meinel band emission, is a prominent feature in night sky spectra. The OH Meinel band emission has been used to monitor atmospheric density changes, temperature fluctuations, and species concentrations for several decades [Meriwether, 1989; Sivjee, 1992; Khomich et al., 2008; Grygalashvyly, 2015].

Collisional relaxation of OH(v) by other atmospheric species plays an important role in determining the observed internal quantum-state distribution. As a result, collisional energy transfer between OH(v) and the major components of the atmosphere at this altitude region, $O_2$ and $N_2$, have been studied for many years. Nevertheless, several gaps persist in our knowledge of these processes. Especially for oxygen atoms, which form a significant component of the atmosphere at the high-altitude part of the OH(v) layer, studies of collisional energy transfer have been relatively limited. Notable recent





developments from laboratory studies include the demonstration that the deactivation of OH(v = 9) by O atoms is characterized by a total loss rate coefficient that is significantly larger than that by $O_2$ and $N_2$, and the most efficient relaxation pathway involves multi-quantum vibrational-to-electronic energy transfer [Kalogerakis et al., 2011; 2016].

The question of whether the OH rotational temperature determined by observations is equivalent to the local kinetic temperature is of fundamental significance and has been debated since the discovery of the Meinel band emission in the 1950s [Kalogerakis et al., 2018; and references therein]. Simultaneous observations of mesospheric OH(v) emissions from several vibrational levels by Cosby and Slanger [2007] and Noll et al. [2015, 2016] using sky spectra from astronomical telescopes reported rotational temperatures that exhibit a clear vibrational level dependence; the rotational temperature increases by approximately 15 K as the OH vibrational quantum number increases from v = 2 to v = 8. Both groups also determined that the rotational temperature of OH(v = 8) was significantly higher than that for OH(v = 9). Figure 1 summarizes the results on OH rotational temperatures reported by Cosby and Slanger [2007], Oliva et al. [2015], and Noll et al. [2016]. Despite some variation, possibly due to the different location, time, and instrument for these measurements, these data sets show a similar trend for the vibrational level dependence of the OH rotational temperatures.

In this Technical Note, we first briefly consider the available knowledge from fundamental theoretical and experimental studies of rotational energy transfer. These studies unambiguously demonstrate that bimodal rotational population distributions are an inherent feature of the rotational relaxation process in gases. Signatures of bimodal behaviour have been observed in the laboratory as well as in the upper atmosphere. We then show that neglecting to account for this bimodality in the mesospheric OH rotational population distributions leads to large systematic errors in the determined rotational temperatures. These findings provide an explanation for the aforementioned dependence of the OH rotational temperatures on the vibrational level determined in previous studies. Finally, this Note briefly discusses the implications for mesospheric temperature measurements and strategies for mitigation of systematic errors.

## 2 Evidence from Studies of Rotational Energy Transfer

Before considering results from atmospheric observations, it is highly informative to review selected information from theoretical studies on the mechanism of rotational relaxation as well as some relevant laboratory results.

In their seminal experiments on rotational energy transfer investigated by the technique of "arrested relaxation" using infrared chemiluminescence, Polanyi and coworkers [Charters and Polanyi, 1962; Anlauf et al., 1967; Polanyi and Woodall, 1972] investigated how the initial highly rotationally excited nonthermal population distribution of hydrogen chloride from the H + $Cl_2$ reaction attained thermal equilibrium in collisions with the bath gas. A general observation in these studies was that rotational energy transfer was less efficient as the rotational excitation increased or, in other words, as the energy spacing between rotational levels became larger. A key finding by Polanyi and coworkers was that rotational-to-translational (R-T) energy transfer of an initial rotationally excited population distribution peaking at high rotational quantum number $J$ does not exhibit a transient peak at intermediate $J$ values. Instead, a bimodal distribution is generated with a peak a high $J$,



reflecting the nascent excited rotational population distribution, as well as a new secondary peak at low *J*, corresponding to the thermal distribution of the bath gas. As the rotational relaxation process progresses, the amplitude of the excited, nonthermal population distribution decreases while that of the thermalized distribution increases accordingly. Polanyi and Woodall [1972] developed a theoretical model for R-T energy transfer that quantitatively accounted for their experimental observations. According to this model, the transition probability for rotational energy transfer decreases exponentially with the energy gap between the two rotational states involved in the rotational energy exchange. Figure 2 demonstrates the bimodal pattern observed when an initial rotational population distribution that is a delta function relaxes according to the model of Polanyi and Woodall [1972]. Exponential-gap models have been extensively used in studies of rotational energy transfer for decades [Koszykowski et al., 1985; Lucht et al, 1986; Dodd et al., 1994; Holtzclaw et al., 1997; Beaud et al. 1998; Fei et al. 1998; Kliner and Farrow, 1999; Hickson et al., 2002; Knopp et al., 2003; Funke et al., 2012, Noll et al.; 2018]

Regarding rotational relaxation involving the hydroxyl radical, Kliner and Farrow (1999) performed relevant, laser-based experiments studying energy transfer in OH(v = 0) excited to rotational levels $N$ = 1-12 near room temperature. In those studies, pulsed photolysis of $H_2O_2$ at 266 nm created a rotationally excited population distribution, whose temporal evolution was probed using laser-induced fluorescence (LIF). Kliner and Farrow (1999) were able to determine that rotational relaxation by $O_2$ and $N_2$ is more efficient for lower rotational levels than for higher ones. They also found that an exponential gap model similar to that of Polanyi and Woodall [1972] reproduced their laboratory measurements remarkably well. Figure 3 presents the results of Kliner and Farrow (1999) for rotationally excited OH(v = 0, N = 1-12) colliding with $N_2$ bath gas. The figure also shows Boltzmann fits to the data of Kliner and Farrow using a fitting function described by two Boltzmann distributions, at low and high temperatures. The initially excited and the final, thermalized distributions were fit to a single-temperature, indicated in Fig. 3. For the other measurements, we constrained the two determined temperatures as fixed values and varied the partitioning of the two rotational level populations at the two characteristic temperatures so as to reflect the changes in the degree of thermal equilibration. As Fig. 3 shows, this experimental system is well described by a low temperature value near room temperature and a high temperature value (294 ± 4 K and 1567 ± 38 K, based on our fits) reflecting the nascent rotational distribution of OH(v = 0) following photodissociation of $H_2O_2$. Gericke et al. [1986] performed a relevant laboratory study investigating the dynamics of $H_2O_2$ photodissociation at 266 nm and found that the nascent OH(v = 0) product rotational state distribution was characterized by a temperature of 1530 ± 150 K , in excellent agreement with the results of the fits shown in Fig. 3. We find similar agreement with the measurements of Kliner and Farrow [1999] for collider gases $O_2$ and Ar. In summary, the results of Kliner and Farrow (1999) provide further validation for the exponential-gap rotational relaxation model of Polanyi and Woodall [1972] for OH as well as a clear laboratory demonstration of bimodality in the OH product state distributions following rotational relaxation.



## 3 Bimodality in Rotational Population Distributions of Mesospheric OH and Implications for Rotational Temperature Measurements

Atmospheric observations have revealed that the rotational population distributions of mesospheric OH display a bimodal character. Early observations provided the first indications for emission from lines associated with high rotational excitation in selected vibrational levels [Pendleton et al., 1989; 1992; Perminov and Semenov, 1992; Smith et al., 1992; Dodd et al., 1993; Perminov et al., 2007; and references therein]. Recent simultaneous observations of multiple OH vibrational levels using high-resolution spectrographs from astronomical telescopes by Cosby and Slanger [2007] and Oliva et al. [2015] represent the most comprehensive demonstrations of bimodal behaviour in OH(v) rotational population distributions to date. Bimodal behaviour is evident for all observed vibrational levels OH(v = 2-9), but this effect may appear at first less pronounced for the highest vibrational levels,. In fact, the opposite is true because the higher the OH vibrational level is, the larger the fraction of the rotational level population that deviates from thermodynamic equilibrium. This behaviour results from the fact that the OH(v) radiative lifetime decreases as the vibrational level increases and, consequently, the higher OH vibrational levels experience fewer collisions with the ambient atmosphere. In principle, more complex behaviour than bimodal might be possible because of the large number of production and removal pathways for mesospheric OH(v). Hints of additional features may be discerned for v = 3-5 in the data of Oliva et al. [2015], but these are at best tentative given the signal-to-noise ratio. Additional measurements at high-resolution and sensitivity combined with careful corrections for any absorption and spectral interferences will be required to settle this question. Based on the available information to date, it appears that to a first approximation the simplest adequate description of the mesospheric OH(v) rotational population distributions is that of bimodal Boltzmann distributions.

We now consider the effect that the presence of a bimodal OH rotational population distributions has on the determination of OH rotational temperatures by considering an example for OH(v = 9). This is the highest populated vibrational level and most probable product of the $H + O_3$ reaction. Collisional cascade from higher vibrational levels can be assumed to be a limited, and likely negligible, source. Therefore, more than any other OH vibrational level, rotational relaxation of v = 9 is expected to follow the exponential-gap model of Polanyi and Woodall [1972]. Figure 4 presents the observed rotational population distribution reported by Noll et al. [2018] for v =9 together with fits we performed using one simple and one bimodal Boltzmann distribution functions. In the former case, only rotational lines with energy less than 250 cm$^{-1}$ are considered. From Fig. 4, we conclude that neglecting the bimodal behaviour of the rotational population distributions and considering only a few rotational lines involving the lowest quantum numbers leads to unacceptably large systematic errors in the extracted OH rotational temperatures. The lower temperature value for the bulk of the population obtained from the fit using a two-temperature, bimodal Boltzmann distribution is 20 K smaller than the temperature obtained using a single Boltzmann function and only a few low-level rotational transitions.

We recently considered two-temperature fits for selected OH vibrational bands from the Oliva et al. data set [Kalogerakis et al., 2018]. The OH(v) rotational temperatures inferred from single and two-temperature Boltzmann distribution functions are generally different. Based on the information above, it becomes clear the observed trend for single-temperature fits does not





reflect real temperature changes; it is an artefact that arises from neglecting the bimodal character of the OH rotational population distribution. Although this effect is most pronounced for the largest OH(v) levels, e.g., v = 8, 9, differences for the lowest observed vibrational levels, v = 2, 3, appear to be comparable to the estimated uncertainties. The majority of the OH(v) product from the H + O$_3$ reaction is generated in the highly vibrationally excited levels v = 7-9, while collisional or

radiative relaxation is needed to generate the lowest vibrational levels. It is reasonable to expect that the lower vibrational levels have undergone more extensive thermalization. At the same time, we do not fully understand all the relevant collisional relaxation processes and the variability of the bimodal character in the OH rotational population distributions. Thus, although we could state with confidence that not accounting for the bimodality in the rotational population distributions introduces large systematic errors for the highest OH vibrational levels, it is presently difficult to assess the

extent to which changes in the rotational temperatures for OH(low-v) are influenced by variations in the fraction of the rotational population distribution that is not in thermal equilibrium.

The most important finding of this Technical Note is the demonstration that the traditional approach in aeronomy to determine OH rotational temperatures using only a pair or a few rotational lines involving the lowest rotational quantum levels does not account for the bimodality of the observed mesospheric OH rotational population distributions and can lead

to unacceptably large systematic errors in the OH rotational temperature determination, especially for OH(high v). To mitigate this problem, the recommended approach would be to concurrently obtain information on the non-equilibrated, high-rotational level tail of the OH(v) rotational population distribution. The adequate resolution and sensitivity to record the full rotational population distribution may not always be available, but even establishing a lower limit for the ratio of the high-rotational level population versus the low-rotational level population would be helpful in assessing potential systematic

errors. Without this type of information, it is not clear what portion of the observed variability in the OH rotational temperature of any specific vibrational level could be attributed to changes in the non-thermalized rotational population.

## 4 Summary and Conclusions

Evidence from theoretical calculations and laboratory experiments demonstrates that rotational energy transfer between a Boltzmann distribution of rotationally excited molecules and a thermal bath leads to bimodal distributions. Such behaviour

has indeed been reported in atmospheric observations of mesospheric OH. The common approach in aeronomy of considering only a few OH rotational lines with the lowest rotational excitation to determine the rotational temperature does not account for the bimodality of the OH(v) rotational population distributions and can lead to large systematic errors overestimating the rotational temperature. These errors are largest for the highest OH(v) vibrational levels and their magnitude can reach several degrees Kelvin. This effect provides an explanation for the apparent vibrational-level

dependence of OH rotational temperatures reported from previous atmospheric observations. Careful consideration of the highly rotationally excited portion of the rotational population distributions under study is required for a reliable determination of rotational temperatures from mesospheric OH(v) Meinel band observations.





*Data Availability.* The data set from Oliva et al. (2015) presented here and information relevant to the analysis is available on the Open Science Framework website https://doi.org/10.17605/OSF.IO/NKWPJ.

*Competing Interests.* The author declares no competing interests.

**Acknowledgements**

The material presented here is based in part on work supported by the US National Science Foundation (NSF) Grant AGS-1441896, NASA grant 80NSSC17K0638, and SRI International R&D funds. The author thanks Philip C. Cosby, Tom G. Slanger, and Stefan Noll for helpful discussions.

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



FIGURES

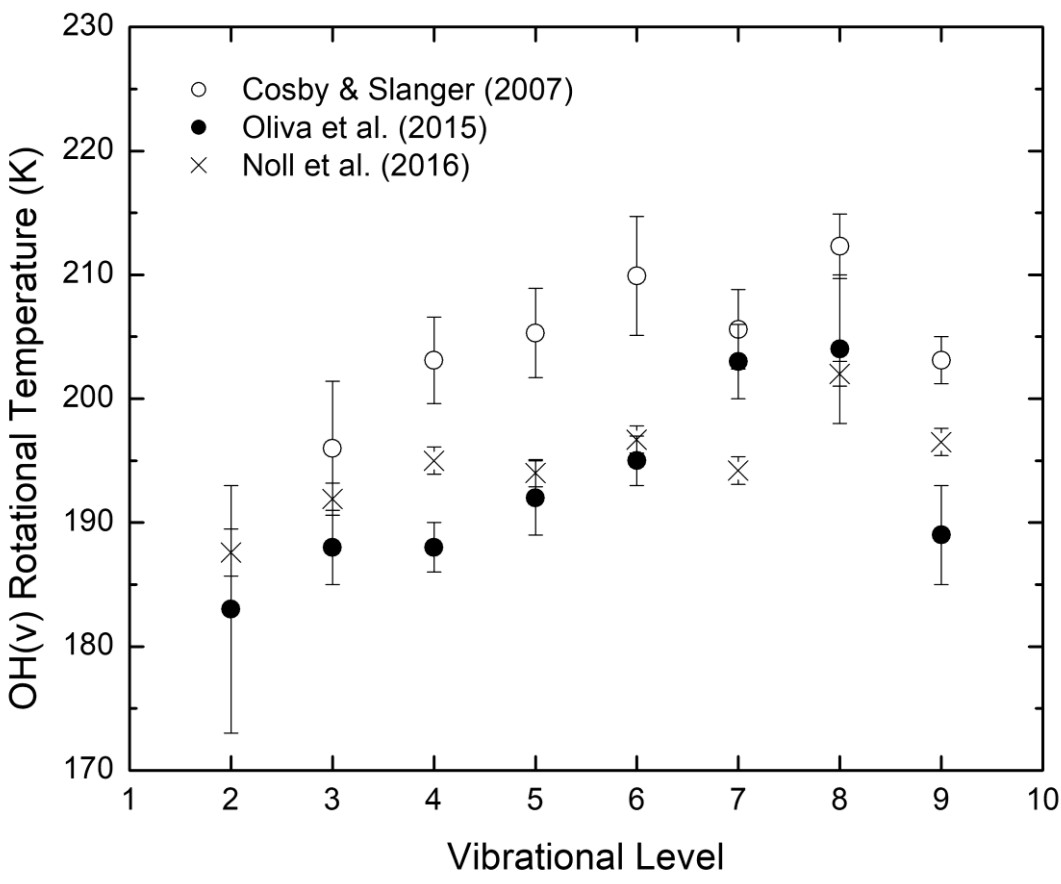

**Figure 1: The OH(v) rotational temperature dependence on the vibrational level v, as reported by recent simultaneous observations of multiple vibrational levels using high-resolution astronomical instruments.**





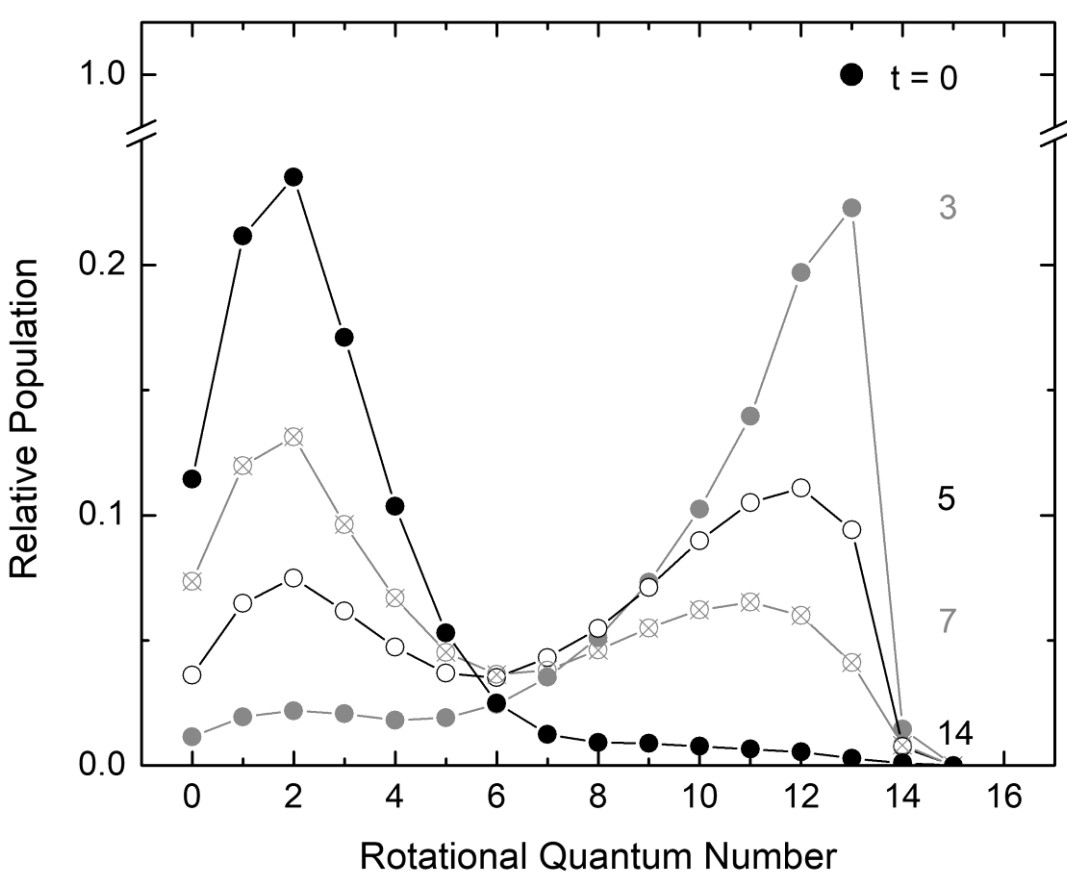

**Figure 2: Temporal evolution of a delta function initial rotational population distribution relaxing in a bath gas according to an exponential-gap model with unrestricted $\Delta J$. Adapted from Fig. 4 of Polanyi and Woodall [1972]. The alternating black and grey labels indicate reduced time units.**





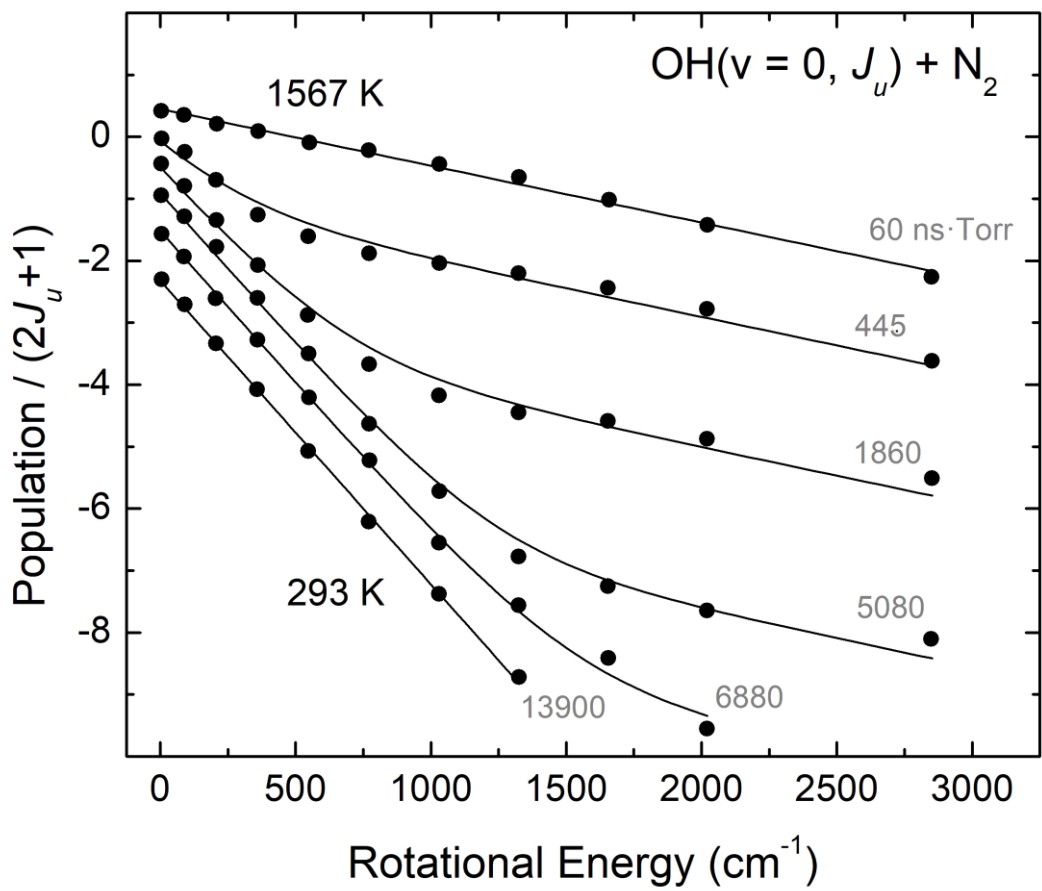

**Figure 3: Experimental results (circles) of Kliner and Farrow (1999) for rotational relaxation of OH(v = 0, N = 1-12) colliding with N₂ bath gas and fits to single-temperature and two-temperature Boltzmann distribution functions. The two characteristic temperatures represent the initial OH distributions and the final fully thermalized gas. The relative weight of the two Boltzmann distributions changes as the relaxation process evolves in time. The grey labels show the product time × pressure corresponding to the experimental measurements.**





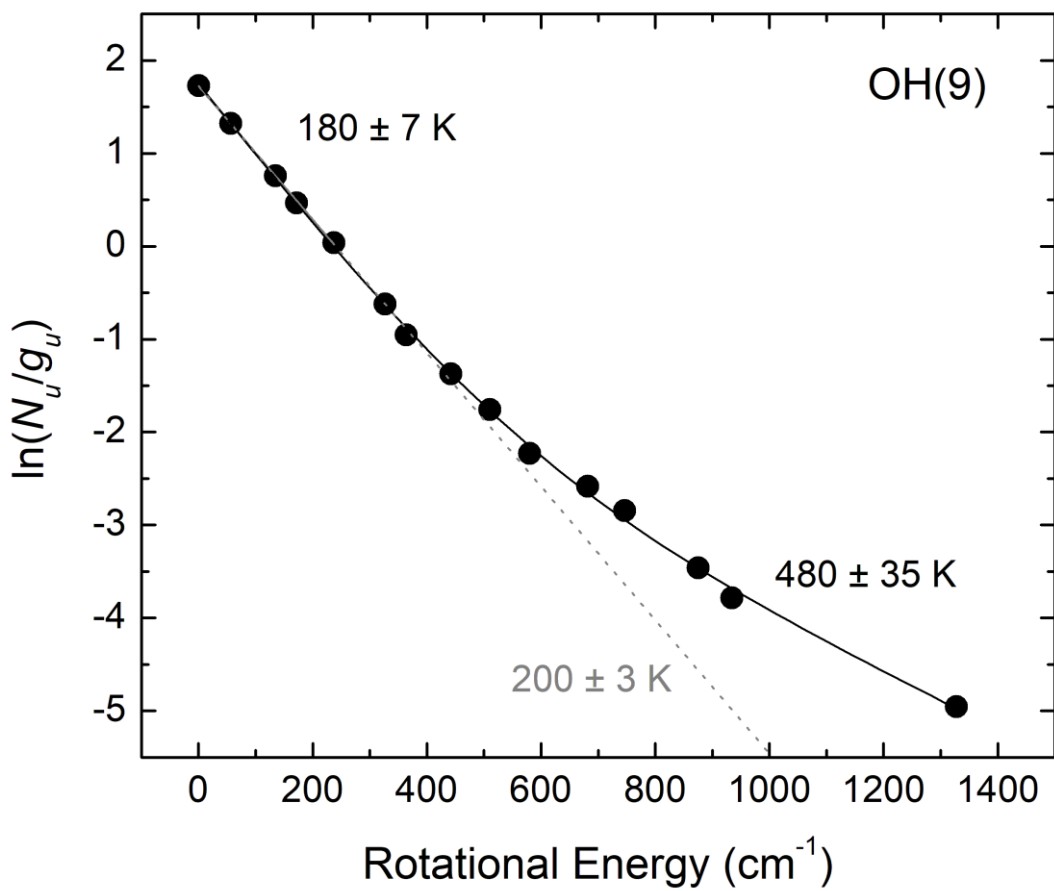

**Figure 4: Mesospheric OH(v = 9) rotational population distribution based on the observations of Noll et al. [2018; Figure 3b]. The dotted grey line shows the result of a single-temperature fit for E < 250 cm⁻¹. The black solid line shows a two-temperature fit using all the data points. Not considering the bimodality of the rotational population results in large systematic errors because the contributions of the non-thermalized Boltzmann distribution to the low rotational energy region are not accounted for.**