# Peer review of "Technical Note: Bimodality in Mesospheric OH Rotational Population Distributions and Implications for Temperature Measurements"

_Atmospheric Chemistry and Physics, 2018_

## Referee Comment (RC1) · Anonymous Referee #1 · 14 Dec 2018

Referee report on manuscript acp-2018-1047

Technical Note: Bimodality in Mesospheric OH Rotational Population Distributions and Implications for Temperature Measurements

By K.S.Kalogerakis

General Comments

1. OH rotational spectra are widely used for the estimation of upper mesosphere temperatures. The author discusses the complexity of the vibrational-rotational structures.

[Figure]

He demonstrates the validity of an exponential-gap rotational relaxation model and the bimodality of the OH state distribution. He suggests that more complicated structures beyond bimodality are possible. These are important and interesting results. 2. The author shows that it is important to take bimodality into account when deriving temperatures from the OH spectrum. Large errrors can occur if bimodality is neglected. This is an important and intriguing result for atmospheric physics. 3. The author makes suggestions how to mitigate the problem. This again is interesting, but not easy to accomplish. 4. The paper is well written. 5. The paper is recommended for publication after minor changes have been made.

Specific Comments.

1. Page 4, Line 12: "…OH radiative lifetime decreases as the vibrational level decrease…" It should be the other way round! Please check!

Technical Corrections

1. Page 3, Line 24, and Fig.3: 294 K or 293 K? 2. Page 5, Line 5: "Thus it is reasonable…" 3. Page 6, Lines 11pp: Anlauf et al. goes after Adler-Golden 4. Page 7, Lines 4pp: Hickson et al. goes after Grygalashviyly

---

## Referee Comment (RC2) · Anonymous Referee #2 · 6 Jan 2019

Review of the Technical Note: "Bimodality in Mesospheric OH Rotational Population Distributions and Implications for Temperature Measurements" by Konstantinos S. Kalogerakis.

General comments.

The paper is devoted to explanation of the OH* rotational temperatures dependence on vibrational numbers. Author found that the existence of bimodal OH* rotational population distributions is an inherent feature of rotational relaxation. In the manuscript

OH* rotational temperatures dependence on vibrational numbers is explained by the bimodality of the OH*(v) rotational population distributions. The result is obtained based on analysis of selected examples from former investigations. On my opinion the provided analysis is correct and author's conclusions are reasonable.

Specific comments.

Is the explanation of the temperature trend by bimodal rotational population distribution only one possible? If – not, please discuss other with corresponding references.

Page 4, line 12: "the fact that the OH(v) radiative lifetime decreases as the vibrational level increases" – add reference.

Technical corrections

I recommend for references in the manuscript to use unique style, i.e. ( ) or [ ] through the entire manuscript

Page 1, line 20: "von Savigny 2017" - add comma.

Page 2, line 26: "[Charters and Polanyi, 1962;" – please add into reference list.

Page 3, line 9: "Lucht et al, 1986" - point after al.

Page 3, line 10: "Fei et al. 1998" – al., "Funke et al., 2012," – 2012; " Noll et al.;" – al.,

Page 3, line 19: "data of Kliner and Farrow" – add year.

Page 4, line 10: "highest vibrational levels,." – without the comma.

Page 4, line 20: "of a bimodal OH" – the (?! I am not sure).

Page 4, line 33: "Oliva et al. data" – add year.

Generally, after specific and technical corrections, I recommend this paper for publication in Atmospheric Chemistry and Physics.

---

## Author Comment (AC1) · 12 Jan 2019

**Response to Comments by Referee #1**

**Referee comments in boldface type**
*Author responses in italics*

*The author sincerely appreciates the referee's offer to review the manuscript, all the associated efforts, and helpful comments.*

**Anonymous Referee #1**
**Referee report on manuscript acp-2018-1047**
**General Comments**
**1. OH rotational spectra are widely used for the estimation of upper mesosphere temperatures. The author discusses the complexity of the vibrational-rotational structures. He demonstrates the validity of an exponential-gap rotational relaxation model and the bimodality of the OH state distribution. He suggests that more complicated structures beyond bimodality are possible. These are important and interesting results.**
**2. The author shows that it is important to take bimodality into account when deriving temperatures from the OH spectrum. Large errrors can occur if bimodality is neglected. This is an important and intriguing result for atmospheric physics.**
**3. The author makes suggestions how to mitigate the problem. This again is interesting, but not easy to accomplish.**
**4. The paper is well written.**
**5. The paper is recommended for publication after minor changes have been made.**

**Specific Comments.**
**1. Page 4, Line 12: ": : :OH radiative lifetime decreases as the vibrational level decrease: : :" It should be the other way round! Please check!**
*The reviewer is correct. This inadvertent mistake had already been corrected in the version submitted to ACP for Discussion and no further action was taken.*

**Technical Corrections**
**1. Page 3, Line 24, and Fig.3: 294 K or 293 K?** *Changed to the correct* $293 \pm 4\ K$
**2. Page 5, Line 5: "Thus it is reasonable: : :"** *Changed as suggested*
**3. Page 6, Lines 11pp: Anlauf et al. goes after Adler-Golden** *Corrected as suggested*
**4. Page 7, Lines 4pp: Hickson et al. goes after Grygalashviyly** *Corrected as suggested*

---

## Author Comment (AC2) · 12 Jan 2019

**Response to Comments by Referee #2**

**Referee comments in boldface type**
*Author responses in italics*

*The author sincerely appreciates the referee's offer to review the manuscript, all the associated efforts, and helpful comments.*

**Anonymous Referee #2**
**Referee report on manuscript acp-2018-1047**
**General comments.**
**The paper is devoted to explanation of the OH\* rotational temperatures dependence on vibrational numbers. Author found that the existence of bimodal OH\* rotational population distributions is an inherent feature of rotational relaxation. In the manuscript OH\* rotational temperatures dependence on vibrational numbers is explained by the bimodality of the OH\*(v) rotational population distributions. The result is obtained based on analysis of selected examples from former investigations. On my opinion the provided analysis is correct and author's conclusions are reasonable.**

**Specific comments.**
**Is the explanation of the temperature trend by bimodal rotational population distribution only one possible? If – not, please discuss other with corresponding references.**
*This report demonstrates that the OH rotational temperature dependence on the vibrational level that has been reported previously based on measurements of only the few lowest rotational levels contains large systematic errors. Therefore, the previously reported trend should be considered an artefact and does not reflect how the temperature of the thermalized portion of the OH rotational population distribution depends on the vibrational level. Not enough information is available yet to establish what the actual trend and its variability are.*
*Another key point is that we do not know how collisional relaxation influences the bimodality in the rotational population distributions. As stated in lines 6-8 of the discussion on page 5, "…we do not fully understand all the relevant collisional relaxation processes and the variability of the bimodal character in the OH rotational population distributions." It seems quite remarkable that after seven decades of measurements on OH rotational temperature, one finds that in many crucial aspects we are just at the beginning.*
*Other than the comments provided above, the manuscript was left unchanged.*

**Page 4, line 12: "the fact that the OH(v) radiative lifetime decreases as the vibrational level increases" – add reference.**
*This sentence was corrected as follows and a reference was added:*
*"This behaviour results from the fact that the OH(v) radiative lifetime decreases as the vibrational level increases (Brooke et al., 2016) and, consequently, the higher OH vibrational levels experience fewer collisions with the ambient atmosphere."*

*Brooke, J. S. A., Bernath, P. F., Western, C. M., Sneden, C., Afşar, M., Li, G., and Gordon, I. E.: Line strengths of rovibrational and rotational transitions in the $X^2\Pi$ ground state of OH, J. Quant. Spectrosc. Ra., 168, 142–157, 2016.*

**Technical corrections**

**I recommend for references in the manuscript to use unique style, i.e. ( ) or [ ] through the entire manuscript**
*Corrected as suggested using parentheses.*

**Page 1, line 20: "von Savigny 2017" - add comma.**
*Corrected as suggested*

**Page 2, line 26: "[Charters and Polanyi, 1962;" – please add into reference list.**
*The following reference was added:*
*Charters, P. E. and Polanyi, J. C.: Energy distribution among reaction products. Part 1—The reaction atomic hydrogen plus molecular chlorine, Discuss. Faraday Soc., 33, 107-117, 1962.*

**Page 3, line 9: "Lucht et al, 1986" - point after al.**
*Corrected as suggested*

**Page 3, line 10: "Fei et al. 1998" – al., "Funke et al., 2012," – 2012; " Noll et al.;" – al.,**
*Corrected as suggested*

**Page 3, line 19: "data of Kliner and Farrow" – add year.**
*Corrected as suggested*

**Page 4, line 10: "highest vibrational levels,." – without the comma.**
*Corrected as suggested*

**Page 4, line 20: "of a bimodal OH" – the (?! I am not sure).**
*This sentence was changed as follows:*
*"We now consider the effect of bimodal OH rotational population distributions on the determination of OH rotational temperatures by considering an example for OH(v = 9)."*

**Page 4, line 33: "Oliva et al. data" – add year.**
*Corrected as suggested*

**Generally, after specific and technical corrections, I recommend this paper for publication in Atmospheric Chemistry and Physics.**